# MARGINALIZATION CONSISTENT MIXTURE OF SEPARABLE FLOWS FOR PROBABILISTIC IRREGULAR TIME SERIES FORECASTING

## ABSTRACT

Probabilistic forecasting models for joint distributions of targets in irregular time series are a heavily under-researched area in machine learning with, to the best of our knowledge, only three models researched so far: GPR, the Gaussian Process Regression model (Dürichen et al., 2015), TACTiS, the Transformer-Attentional Copulas for Time Series (Drouin et al., 2022; Ashok et al., 2024) and ProFITi (Yalavarthi et al., 2024b), a multivariate normalizing flow model based on invertible attention layers. While ProFITi, thanks to using multivariate normalizing flows, is the more expressive model with a better predictive performance, we will show that it suffers from marginalization inconsistency: it does not guarantee that the marginal distributions of a subset of variables in its predictive distributions coincide with the directly predicted distributions of these variables. Also, TACTiS does not provide any guarantees for marginalization consistency.

We develop a novel probabilistic irregular time series forecasting model, Marginalization Consistent Mixtures of Separable Flows (`moses`), that mixes several normalizing flows with (i) Gaussian Processes with full covariance matrix as source distributions and (ii) a separable invertible transformation, aiming to combine the expressivity of normalizing flows with the marginalization consistency of Gaussians. In experiments on four different datasets we show that `moses` outperform other state-of-the-art marginalization consistent models, perform on par with ProFITi, but different from ProFITi, guarantees marginalization consistency.

## 1 INTRODUCTION

In many real-world domains ranging from health to astronomy time-variant data is measured in an irregular fashion: different channels are measured at different times and usually not on a regular grid. Besides mere point estimates, i.e., just the **expected target values** for some channels at some future time points, one is usually interested in a probabilistic/distributional forecast of **a distribution of target values**. A full predicted distribution of target values provides way more information about the targets. For example, when bioengineers are growing bacteria in tanks they are interested in predicting the oxygen levels. The expected value is already interesting, but if additionally we can predict more fine-grained the whole distribution of possible oxygen levels this allows them now to quantify the risk that the oxygen level falls below some critical threshold and all bacteria die; the expected value alone is not sufficient to do so. Consequently, in recent years several models have been developed for probabilistic forecasting of irregular time series (De Brouwer et al., 2019; Deng et al., 2020; Biloš et al., 2021; Schirmer et al., 2022).

However, all these models are limited to forecasting the distribution of the **value of a single channel at a single time**. But decision makers often require information about several variables and their interaction, that is, about the predicted joint distribution of several channels at several times. In our initial example, the bioengineers will be interested not just in the forecasts of the oxygen levels and the biomass, but also in their interaction: a risk for either low biomass or low oxygen levels might be tolerable, but a risk that both occur at the same time will require their intervention.

Probabilistic forecasting models for joint distributions of targets in irregular time series are a heavily under-researched area in machine learning with, to the best of our knowledge, only three models

researched so far: GPR, the Gaussian Process Regression model (Dürichen et al., 2015) uses a multivariate normal distribution as predictive distribution, TACTiS, the Transformer-Attentional Copulas for Time Series (Drouin et al., 2022; Ashok et al., 2024) uses a copula with normalizing flows for the marginal distributions, and ProFITi (Yalavarthi et al., 2024b) uses a multivariate normalizing flow model based on invertible attention layers. ProFITi, thanks to using normalizing flows, is able to learn way more expressive predictive distributions, for example, it could express multi-modal predictive distributions, while GPR being limited to multivariate Gaussians cannot do that. Consequently, it outperforms GPR by a wide margin in experiments.

Models for probabilistic forecasting of joint target distributions in irregular time series have to be able to express distributions over a varying number of variables, as by the nature of irregular time series, the number of observations that fall into a given span of time might vary. However, the capability to provide predictive distributions for a varying number of observations/variables introduces a new issue: to get the predictive joint distribution for values of channels at some time points, we can query the model multiple ways: (i) we can just ask for the predictive distribution of those variables directly or (ii) we can ask for the joint predictive distribution of all variables, and then marginalize out the variables that were not queried. Using a consistent model we expect to get the same answer either way — we call this property **marginalization consistency**. An important special case is that the univariate marginals of the models' joint predictive distributions agree with the univariate predictive distributions it provides when directly queried.

We show that the currently best performing probabilistic forecasting model for joint target distributions in irregular time series, ProFITi, neither provides any guarantees to be marginalization consistent nor empirically happens to be marginalization consistent on the standard datasets used in experiments. Its main competitor, TACTiS, does not provide guarantees for marginalization consistency either. In contrast, while GPR is marginalization consistent, it shows worse predictive performance.

From this starting point we will construct a novel model that combines the ideas of Gaussian Processes and normalizing flows in a way completely different from ProFITi and GPR, to achieve both, guaranteed marginalization consistency and high predictive accuracy (see Figure 4).

Overall our contributions as follows:

1. We propose a measure for the degree of marginalization consistency of models for joint distributions with varying size, the Wasserstein Distance between the (possibly numerically) marginalized predicted joint distribution of several variables and the directly predicted marginal distribution (Section 6);

2. We show that the currently best performing model for probabilistic irregular time series forecasting, ProFITi, does not provide any guarantees for marginalization consistency, and in experiments we show furthermore that it actually suffers from marginalization inconsistency (Section 7);

3. We develop a novel probabilistic irregular time series forecasting model, Marginalization Consistent Mixtures of Conditional Flows (moses), that mixes several normalizing flows with (i) Gaussian Processes **with full covariance matrix** as source distributions (but the usual identity matrix) and (ii) a **separable** invertible transformation (i.e., for each dimension separately, instead of the usual multivariate ones), aiming to combine the expressivity of normalizing flows with the marginalization consistency of mixtures of Gaussian Processes (Sections 4 and 5);

4. We prove that Marginalization Consistent Mixtures of Separable Flows are guaranteed marginalization consistent (Sections 4 and 5);

5. In experiments on four different datasets we show that Marginalization Consistent Mixtures of Separable Flows outperform other state-of-the-art marginalization consistent models, perform on par with ProFITi, but guarantee marginalization consistency (Section 7);[1]

---

[1] Code available at https://anonymous.4open.science/r/seperable_flows-BACC

## 2 PRELIMINARIES

We make use of the triplet representation of an irregular time series for probabilistic forecasting (Horn et al., 2020). Here, an **irregularly sampled time series** $X$ is a sequence of $N$-many triplets:

$$X := \left( (t_n^{\text{OBS}}, c_n^{\text{OBS}}, v_n^{\text{OBS}}) \right)_{n=1:N} \in \text{Seq}(\mathcal{X}) \qquad \mathcal{X} = \mathbb{R} \times \{1, \ldots, C\} \times \mathbb{R} \tag{1}$$

where $t_n^{\text{OBS}} \in \mathbb{R}$ is the observation time point, and $v_n^{\text{OBS}} \in \mathbb{R}$ is the observed value in channel $c_n^{\text{OBS}} \in \{1, \ldots, C\}$. A **time series query** $Q$ is a sequence of $K$-many pairs:

$$Q := \left( (t_k^{\text{QRY}}, c_k^{\text{QRY}}) \right)_{k=1:K} \in \text{Seq}(\mathcal{Q}) \qquad \mathcal{Q} = \mathbb{R} \times \{1, \ldots, C\} \tag{2}$$

where $t_k^{\text{QRY}} \in \mathbb{R}$ is the future time point and $c_k^{\text{QRY}} \in \{1, \ldots, C\}$ is the queried channel. A **forecasting answer** $y$ is a sequence of scalars: $y = (y_1, \ldots, y_K)$, where $y_k$ is the forecasted value in channel $c_k^{\text{QRY}}$ at time $t_k^{\text{QRY}}$. Here, $\text{Seq}(\mathcal{X})$ denotes the space of finite sequences over $\mathcal{X}$.

For forecasting, all the query time points are after the observations: $\min_{k=1:K} t_k^{\text{QRY}} > \max_{n=1:N} t_n^{\text{OBS}}$. The task of probabilistic irregular time series forecasting is to find a model $\hat{p}$ that can predict the joint multivariate distribution $\hat{p}(y \mid Q, X)$ of the answers $y$, given the query points $Q$ and observed series $X$. Both the context $N = |X|$ and the query length $K = |Q|$ are allowed to be dynamic (**R1**).

As the time stamp and channel-ID are included in each sample, the order of the samples does not matter, and hence any model prediction should be independent of it (**R2**). Moreover, for a subquery $Q'$ of $Q$, there are two different ways to predict the joint distribution: either by marginalizing the predicted distribution of the complete query $Q$, or by predicting the joint distribution of the subquery $Q'$ directly. The two predictions should be equivalent (**R3**).

**Requirements.** A marginalization consistent probabilistic irregularly sampled time series forecasting model must satisfy the following requirements:

**R1 Joint Multivariate Prediction**.
The model $\hat{p}$ can predict the joint distribution across multiple time steps of a multivariate time series for arbitrary sizes of both the query $K = |Q|$ and context $N = |X|$.

$$\hat{p}: \text{Seq}(\mathbb{R} \times \mathcal{Q}) \times \text{Seq}(\mathcal{X}) \longrightarrow \mathbb{R}_{\geq 0},$$
$$(y, Q, X) \longmapsto \hat{p}(y_1, \ldots, y_K \mid Q_1, \ldots, Q_K, X_1, \ldots, X_N) \tag{3}$$

So that, for a given pair $(Q, X)$, the partial function $(y_1, \ldots, y_K) \mapsto \hat{p}(y_1, \ldots, y_K \mid Q, X)$ realizes a probability density on $\mathbb{R}^{|Q|}$.

**R2 Permutation Invariance**.
The predicted density should be invariant under permutations of both the query or context:

$$\hat{p}(y \mid Q, X) = \hat{p}(y^\pi \mid Q^\pi, X^\tau) \qquad \forall \pi \in S_{|Q|}, \tau \in S_{|X|} \tag{4}$$

**R3 Marginalization Consistency/Projection Invariance**.
Predicting the joint density for the sub-query $Q_{-k}$ given by removing the $k$-th item should yield the same result as marginalizing the $k$-th variable from the complete query.

$$\hat{p}(y_{-k} \mid Q_{-k}, X) = \int_{\mathbb{R}} \hat{p}(y \mid Q, X) \, dy_k \tag{5}$$

This generalizes to any subset $K_S \subseteq \{1, \ldots, K\}$.

For a model satisfying **R1**-**R3**, we will only have to marginalize if we try to validate the marginalization consistency. For this validation we added requirement **R3**. Yalavarthi et al. (2024b) discussed **R1** and **R2**, but did not consider **R3**. We argue that irregularly sampled time series is realization of a stochastic process and **R3** is a fundamental property of any model that mimics it.

**Theorem 1.** *Any model that satisfies **R1**-**R3** realizes an $\mathbb{R}$-valued stochastic process over the index set $T = \mathbb{R} \times \{1, \ldots, C\}$.*
*Proof. This is a direct application of Kolmogorov's extension theorem (Øksendal, 2003)*

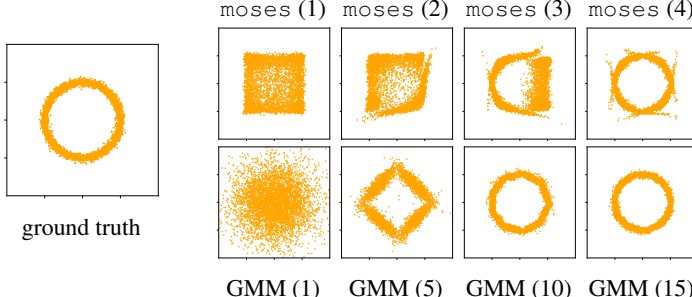

Figure 1: (Top) Importance of multiple flow components: moses(1) cannot represent the correct distribution, but moses(4) can. (Bottom) Limitation of Gaussian Mixture Models: GMM needs 15 components to match the distribution of moses(4).

## 3 Related Work

There have been multiple works that deal with point forecasting of irregular time series (Ansari et al., 2023; Che et al., 2018; Chen et al., 2024; Yalavarthi et al., 2024a). In this work we deal with probabilistic forecasting of irregular time series. Models such as NeuralFlows (Biloš et al., 2021), GRU-ODE (De Brouwer et al., 2019), and CRU (Schirmer et al., 2022) predict only the marginal distribution for a single time stamp. Additionally, interpolation models like HetVAE (Shukla & Marlin, 2022) and Tripletformer (Yalavarthi et al., 2023) can also be applied for probabilistic forecasting. However, they also produce only marginal distributions. All the above models assume underlying distribution is Gaussian which is not the case for lots of real-world datasets. On the other hand, Gaussian Process Regression (Dürichen et al., 2015), TACTiS/TACTiS-2, the Transformer-Attentional Copulas for Time Series (Drouin et al., 2022; Ashok et al., 2024) and ProFITi (Yalavarthi et al., 2024b) can predict proper joint distributions. TACTiS is a copula model and ProFITi a conditional normalizing flow model, both can predict any arbitrary distribution. But as both use self-attention on the queries in their encoders, their parametrization of their copula and flow, respectively, depends on all queries in a complex way, and thus they cannot provide any marginalization guarantees.

There have been works on models for tractable and consistent marginals for fixed number of variables such as tabular data. Probabilistic Circuits (Choi et al., 2020) create a sum-prod network on the marginal distributions in such a way that marginals are tractable and consistent. Later, univariate normalizing flows were added to the leaf nodes of the circuit for better expressivity by Sidheekh et al. (2023). However, it is not trivial to extend such circuits to deal with sequential data of variable size. Gaussian Mixture Models (GMMs) (Duda & Hart, 1974) are often used only for unconditional density estimation, but can be extended to conditional density estimation. They can provide tractable and consistent marginal distributions. However, GMMs are not expressive enough and often require a very large number of components to approximate even simple distributions, as shown in Figure 1. Note that normalizing flow models such as (Dinh et al., 2017; Papamakarios et al., 2017; 2021) neither provide tractable marginals nor are applicable to varying number of variables.

Existing works have explored mixtures of normalizing flows for fixed-length sequences. For example, Pires & Figueiredo (2020) and Ciobanu (2021) used flows with affine coupling or masked autoregressive transformations for density estimation, while Postels et al. (2021) applied them to reconstruction tasks. However, these models cannot handle dynamic sequence lengths, and their marginals are intractable.

## 4 Constructing Marginalization Consistent Conditional Distributions

Our goal is to build a model for the conditional joint distribution $p(y_1, \ldots, y_K \mid Q_1, \ldots, Q_k, X)$, as in Equation (3). Since the model should satisfy **R3**, it follows that the marginal distribution of $y_k$ must only depend on $Q_k$ and $X$.

**Separably Parametrized Gaussians.**   The arguably most simple model for a permutation invariant conditional distribution for variably many variables is the family of multivariate Normal distributions $\mathcal{N}(y \mid \mu(x), \Sigma(x))$, whose conditional mean function $\mu(x)$ and conditional covariance function $\Sigma(x)$ are separable, i.e.:

$$\mu_k = \tilde{\mu}(Q_k, X) \qquad\qquad \Sigma_{k,\ell} = \widetilde{\Sigma}(Q_k, Q_\ell, X) \qquad\qquad (6)$$

with mean function $\tilde{\mu}\colon \mathcal{Q} \times \mathrm{Seq}(\mathcal{X}) \to \mathbb{R}$ and a covariance function $\widetilde{\Sigma}\colon \mathcal{Q} \times \mathcal{Q} \times \mathrm{Seq}(\mathcal{X}) \to \mathbb{R}$, a setup very well known from Gaussian processes. Such a separably parametrized multivariate Gaussian is marginalization consistent by design, as marginalizing a Normal distribution boils down to relevant rows and columns of the covariance matrix and the corresponding elements of the mean vector.

However, Gaussian Processes form a restrictive class of models, as any joint distribution of variables is Gaussian. To capture more complex distributions, more expressive models are required. Normalizing flows are a popular choice for this task (Rezende & Mohamed, 2015; Papamakarios et al., 2021).

**Separable Normalizing Flows.**   Normalizing flows model distributions by transforming a source distribution $p_Z$ on $\mathbb{R}^K$ by means of an invertible transformation $f\colon \mathbb{R}^K \to \mathbb{R}^K$. Then the target distribution, the distribution of the image of $f$, can be concisely described by the transformation theorem for densities

$$p_Y(y) \coloneqq p_Z(f^{-1}(y; \theta)) \cdot \left| \det\left( \frac{\partial f^{-1}(y; \theta)}{\partial y} \right) \right| \qquad\qquad (7)$$

Existing approaches to normalizing flows use very simple source distributions, typically a multivariate standard normal $p_Z(z) \coloneqq \mathcal{N}(z \mid 0, \mathbb{I})$, and model interactions between variables by means of the transformation (Rezende & Mohamed, 2015; Papamakarios et al., 2021). Current approaches for conditional normalizing flows for a variadic number of variables followed the same approach and tackled the problem by engineering expressive transformations between vectors of same size, for any size (Liu et al., 2019; Biloš & Günnemann, 2021; Yalavarthi et al., 2024b). For example, ProFITi uses an invertible attention mechanism.

All these models in general will not have a guarantee for marginalization consistency. To the best of our knowledge, there is no simple condition on the transform that would provide such a guarantee.

We therefore propose a drastic change, reversing the standard approach for normalizing flows: to combine (i) simple, separable transforms with (ii) a richer source distribution, namely a Gaussian Process with full covariance matrix. This way interactions between variables cannot be represented by the transformation anymore, but they can be represented by the covariance of the source distribution.

**Lemma 1.** A conditional normalizing flow model over $\mathbb{R}^K$ or $\mathrm{Seq}(\mathbb{R})$ is called *separable*, if it can be expressed in the form

$$f(z \mid Q, X) = (\phi(z_1 \mid Q_1, X), \dots, \phi(z_K \mid Q_K, X)) \qquad\qquad (8)$$

for some univariate function $\phi\colon \mathbb{R} \times \mathcal{Q} \times \mathrm{Seq}(\mathcal{X}) \to \mathbb{R}$, that is invertible in the first argument. Any model that consists of such a separable flow transformation, combined with a marginalization consistent model for the source distribution, is itself marginalization consistent. (Proof: Appendix A.1)

**Conditional Mixtures of Flows.**   When using separably parametrized Gaussians as source distributions in Lemma 1, and expressive univariate transformations, we can model any kind of marginal as well as rich interactions between variables. However, the model is still restricted in its expressiveness, allowing for variable-wise separable transformations of a unimodal (Gaussian) distribution only. We therefore resort to the most simple way to further increase the expressiveness of the model: we combine several of such separable flows into a mixture. Figure 1 shows that even just a few components can lead to a much more expressive model, in particular comparable to a simple GMM without flow transformations.

**Lemma 2.** Given probabilistic models $(\hat{p}_d)_{d=1:D}$ that satisfy **R1**-**R3**, then a mixture model

$$\hat{p}(y \mid Q, X) = \sum_{d=1}^{D} w_d(X)\, \hat{p}_d(y \mid Q, X) \qquad\qquad (9)$$

with permutation invariant weight function $w\colon \mathrm{Seq}(\mathcal{X}) \to \Delta^D$, were $\Delta^D$ denotes probability simplex in $D$ variables: $\Delta^D \coloneqq \{w \in \mathbb{R}^D \mid w_d \geq 0, \sum_d w_d = 1\}$, also satisfies **R1**-**R3**. (Proof: Appendix A.2)

Figure 2: Illustration of proposed `moses`. $D$-many flows (fixed). $K$-many variables (variable). Encoder (enc) takes $X, Q$ (observed series and query timepoint-channel ids.) as input, and outputs an embedding $\mathbf{h}$ (depends on both $X$, and $Q$) and $w$ (depends on $X$ only). $\mu, \Sigma$ of $p_{Z_d}$ are parametrized by $\mathbf{h}_d$. Flow transformation of $p_{Z_d}$ is parametrized by $\mathbf{h}_d$. Transformation layer consists of $K$-many univariate transformations $\phi$ that transforms $z_k$ of $z \sim p_{Z_d}(z \mid \mathbf{h}_D)$ to $y_k$ of $y \sim p_d^{\text{FLOW}}(y \mid \mathbf{h}_d)$.

## 5 MIXTURES OF SEPARABLE FLOWS (MOSES)

Based on the constructions from the last section, we propose to build a marginalization consistent model for forecasting irregular time series in four components (see Figure 2):

1. A separable encoder, consisting of

   (i) A shared encoding $\mathbf{h}^{\text{OBS}} := \text{enc}^{\text{OBS}}(X; \theta^{\text{OBS}})$ of the observations, used for all queries.
   (ii) $D$-many encodings $\mathbf{h}_{d,k} := \text{enc}^{\text{QRY}}(Q_k, X; \theta_d^{\text{QRY}})$ of each query and entire context.

2. $D$-many Gaussian Processes $p_{Z_d}(z \mid \mu_d, \Sigma_d)$, each separably parametrized according to (6), by the encoder for queries $\mathbf{h}_d$.

3. $D$-many separable normalizing flows $\hat{p}_d^{\text{FLOW}}$, one on top of each of the source distributions, whose transformations $f_d$ are also separably parametrized by the encoded queries $\mathbf{h}_d$.

4. A mixture of the $D$-many normalizing flows with mixing weights $w := w(\mathbf{h}^{\text{OBS}})$, depending only on the encoded observations $\mathbf{h}^{\text{OBS}}$, but not the queries.

**1. Separable Encoder.** To encode both the observations $X = ((t_n^{\text{OBS}}, c_n^{\text{OBS}}, v_n^{\text{OBS}}))_{n=1:N}$ and queries $Q = ((t_k^{\text{QRY}}, c_k^{\text{QRY}}))_{k=1:K}$, we apply a positional embedding with learnable parameters $(a_f, b_f)_{f=1:F}$ to the time component (Kazemi et al., 2019).

$$\text{pos\_embed}(t)_f := \begin{cases} a_f t + b_f & \text{if } f = 1 \\ \sin(a_f t + b_f) & \text{else} \end{cases} \tag{10}$$

And one-hot encodings for the channel component. The value is simply passed through.

$$\mathbf{x} := [\text{pos\_embed}(t_n^{\text{OBS}}), \text{one-hot}(c_n^{\text{OBS}}), v_n^{\text{OBS}}]_{n=1:N} \qquad (\in \mathbb{R}^{N \times (F+C+1)}) \tag{11a}$$

$$\mathbf{q} := [\text{pos\_embed}(t_k^{\text{QRY}}), \text{one-hot}(c_k^{\text{QRY}})]_{k=1:K} \qquad (\in \mathbb{R}^{K \times (F+C)}) \tag{11b}$$

The observations are further encoded via self-attention and the queries via cross-attention w.r.t. the encoded observations:

$$\mathbf{h}^{\text{OBS}} := \text{MHA}(\mathbf{x}, \mathbf{x}, \mathbf{x}; \theta^{\text{OBS}}) \qquad (\in \mathbb{R}^{N \times M}) \tag{12a}$$

$$\widetilde{\mathbf{h}} := \text{MHA}(\mathbf{q}, \mathbf{h}^{\text{OBS}}, \mathbf{h}^{\text{OBS}}; \theta^{\text{QRY}}) \qquad (\in \mathbb{R}^{K \times D \cdot M}) \tag{12b}$$

$$\mathbf{h} := \text{reshape}(\widetilde{\mathbf{h}}) \qquad (\in \mathbb{R}^{D \times K \times M}) \tag{12c}$$

where MHA denotes multihead attention. For the encoding of the queries we use an encoding dimension $D \cdot M$ and reshape each $\mathbf{h}_k$ into $D$ encodings $\mathbf{h}_{d,k}$ of dimension $M$.

**2.** $D$ **separably parametrized Gaussian source distributions** $p_{Z_d}(z \mid \mu_d, \Sigma_d)$**.** We model means and covariances simply by a linear and a quadratic function in the encoded queries $\mathbf{h}_d$:

$$\mu(\mathbf{h}_d) = \mathbf{h}_d \theta^{\text{MEAN}} \qquad \implies \qquad \mu(\mathbf{h}_d)_k = \mathbf{h}_{d,k} \theta^{\text{MEAN}} \tag{13a}$$

$$\Sigma(\mathbf{h}_d) = \mathbb{I}_K + \frac{(\mathbf{h}_d \theta^{\text{COV}})(\mathbf{h}_d \theta^{\text{COV}})^T}{\sqrt{M'}} \implies \Sigma(\mathbf{h}_d)_{k,l} = \delta_{kl} + \frac{(\mathbf{h}_{d,k} \theta^{\text{COV}})(\mathbf{h}_{d,l} \theta^{\text{COV}})^T}{\sqrt{M'}} \tag{13b}$$

where $\theta^{\text{MEAN}} \in \mathbb{R}^{M \times 1}$ and $\theta^{\text{COV}} \in \mathbb{R}^{M \times M'}$ are trainable weights shared across all $D$ mixture components. $\mathbb{I}_K$ is an identity matrix of size $K$, and $\delta_{kl} = 1$ if $k = l$ else 0 denotes the Kronecker delta. In Eq. (13b), we divide the inner product with $\sqrt{M'}$ for stable learning, as done in (Vaswani et al., 2017). Since $\Sigma(\mathbf{h}_d)$ is the sum of a positive semi-definite matrix and positive definite matrix, it is guaranteed to be positive definite itself. Note that, $\mathbf{h}_d$, which is encoded from both context $X$ and queries $Q$, takes the role of $X$ and $Q$ together in (6).

**3.** $D$ **separable normalizing flows** $\hat{p}_d^{\text{FLOW}}$**.** To achieve separable invertible transformations, any univariate bijective functions can be applied on each variable separately. Spline based functions attracted interest due to their expressive and generalization capabilities (Durkan et al., 2019; Dolatabadi et al., 2020). We employ computationally efficient Linear Rational Spline (LRS) transformations (Dolatabadi et al., 2020). For a conditional LRS $\phi(z_k; \mathbf{h}_{d,k}, \theta^{\text{FLOW}})$, the function parameters such as width and height of each bin, the derivatives at the knots, and $\lambda$ are computed from the conditioning input $\mathbf{h}_{d,k}$ and some model parameters $\theta^{\text{FLOW}}$. $\theta^{\text{FLOW}}$ helps to project $\mathbf{h}_{d,k}$ to the function parameters, and is common to all the variables $z_{1:K}$ so that the transformation $\phi$ can be applied for varying number of variables $K$. Note that we also share the same $\theta^{\text{FLOW}}$ across all the $D$-many mixture components as well. For details, see Appendix A.4.

**4. Mixture Model.** We model the mixture weights via cross attention, using trainable parameters $\beta \in \mathbb{R}^{D \times M}$ as attention queries, and a softmax to ensure the weights to sum to 1:

$$w := \text{softmax}(\text{MHA}(\beta, \mathbf{h}^{\text{OBS}}, \mathbf{h}^{\text{OBS}}; \theta^{\text{MIX}})) \tag{14}$$

**Theorem 2.** *Our model,* moses*, satisfies* **R1-R3** *and hence realizes a stochastic process via Kolmogorov's Extension Theorem (see Theorem 1). Proof. See Appendix A.3.*

**Training.** Given a batch of training instances $\mathcal{B}$, where for each instance, we have $Q$, $X$ and $y$, we minimize the normalized joint negative log-likelihood (njNLL) (Yalavarthi et al., 2024b):

$$\mathcal{L}^{\text{njNLL}}(\theta) = \frac{1}{|\mathcal{B}|} \sum_{(Q,X,y) \in \mathcal{B}} -\frac{1}{|y|} \log \hat{p}(y \mid Q, X) \tag{15}$$

where $\theta := (\theta^{\text{OBS}}, \theta^{\text{QRY}}, \theta^{\text{MIX}}, \theta^{\text{MEAN}}, \theta^{\text{COV}}, \theta^{\text{FLOW}})$. njNLL generalizes negative log-likelihood to varying number of variables.

# 6 MEASURING MARGINALIZATION CONSISTENCY VIOLATION

To measure the marginalization inconsistency in a probabilistic model, we use the **Wasserstein distance** (WD), also known as **earth mover's distance** (EMD). It quantifies the distance between two probability distributions, representing the "cost" of shifting probability mass from one distribution to another. We consider the Wasserstein distance between the two distributions:

1. The univariate marginals directly predicted with the model.
2. The univariate marginals obtained from numerically integrating the joint distribution.

Corresponding to the left and right-hand side of Equation (5). The Wasserstein $r$-distance between two univariate empirical distributions, $p_U^{\text{EMP}}$ and $p_V^{\text{EMP}}$, given by the samples $U = (u_1, \ldots, u_N)$ and $V = (v_1, \ldots, v_N)$, can be expressed in terms of the order statistics: (Bobkov & Ledoux, 2019)

$$\text{WD}_r(p_U^{\text{EMP}}, p_V^{\text{EMP}}) = \left( \frac{1}{N} \sum_{n=1}^{N} \| u_{\pi_n} - v_{\tau_n} \|^r \right)^{1/r} \tag{16}$$

here $\pi$ and $\tau$ are the permutations that sort the samples $U$ and $V$ respectively.

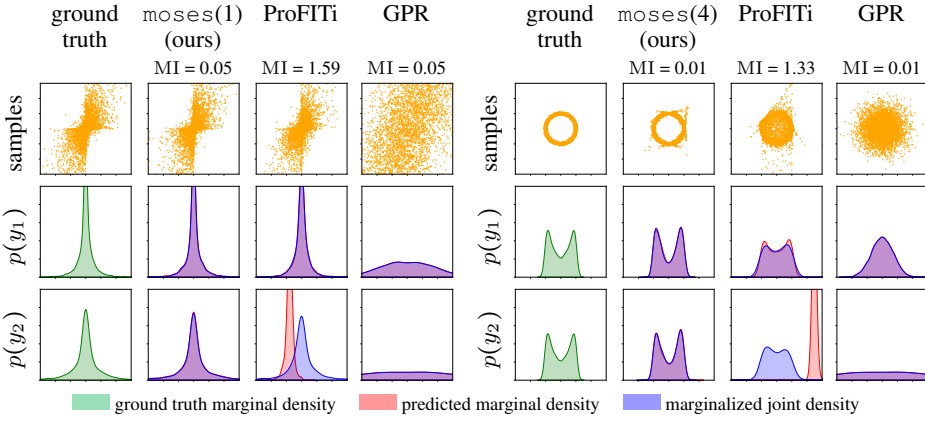

Figure 4: Demonstration of marginal consistency for `moses` (ours), ProFITi Yalavarthi et al. (2024b), and Gaussian Process Regression Bonilla et al. (2007) on two toy datasets: blast (left) and circle (right). ProFITi is inconsistent with respect to the marginals of the second variable $y_2$, while `moses` is consistent with the marginals of both $y_1$ and $y_2$. `moses`($D$) indicates $D$ mixture components. Gaussian Process Regression (GPR) is marginalization consistent but predicts incorrect distributions.

For models $\hat{p}$ that can predict both: (i) The univariate marginals $\hat{p}(y_k \mid Q_k, X)$, for $k = 1, \ldots, K$, (ii) The joint distribution $\hat{p}(y \mid Q, X)$, we can evaluate the **marginalization inconsistency** by comparing the predicted marginals $\hat{p}(y_k \mid Q_k, X)$ to the marginals computed from the joint distribution $\hat{p}^{\text{MAR}}(y_k \mid Q_k, X)$. These computed marginals are obtained by integrating the joint distribution over all variables except $y_k$, i.e.,

$$\hat{p}^{\text{MAR}}(y_k \mid Q_k, X) \coloneqq \int_{\mathbb{R}} \hat{p}(y \mid Q, X) \, \mathrm{d}y_{-k}$$

Sampling from $\hat{p}(y_k \mid Q_k, X)$ is straightforward, but sampling from the marginal $\hat{p}^{\text{MAR}}(y_k \mid Q_k, X)$ is more challenging due to integration. To address this,

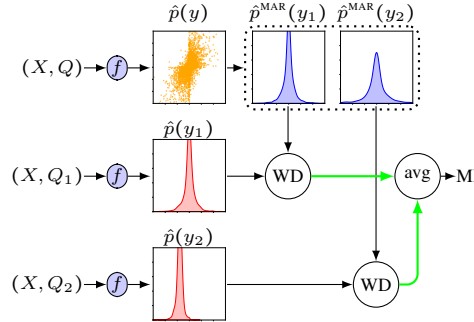

Figure 3: Computing Marginalization Inconsistency of a prediction model. $|Q| = 2$.

we sample from the joint distribution $\hat{p}(y \mid Q, X)$ and treat the $k$-th dimension as a sample from the marginal distribution $\hat{p}^{\text{MAR}}(y_k \mid Q_k, X)$. We demonstrate this in Figure 3. The marginalization inconsistency MI is then defined as the average Wasserstein distance between the predicted marginals and the computed marginals, across all variables $k$:

$$\text{MI}\big(\hat{p}(y \mid X, Q)\big) \coloneqq \frac{1}{K} \sum_{k=1}^{K} \text{WD}_r\big(\hat{p}(y_k \mid Q_k, X), \hat{p}^{\text{MAR}}(y_k \mid Q_k, X)\big) \tag{17}$$

In experiments, we set $r = 2$ and approximate (17) using the Wasserstein distance using 1000 samples.

## 7 EXPERIMENTS

**Toy experiment.** We show that `moses` maintains marginalization consistency using two simple bivariate distributions (Figure 4). The equations to create these distributions are in Appendix B. The goal is to estimate unconditional probability distribution. `moses` accurately predicts both joint and marginal distributions, and also marginalization consistent. ProFITi correctly predicts joint distributions for the blast dataset and nearly correct ones for the circle dataset, but fails in predicting the marginal distribution for the second variable in both cases. This is because ProFITi learn joint distribution in such a way that the second variable always depend on the first. On the other hand, the Gaussian Process Regression model is consistent with marginalization but cannot predict accurately.

**Main experiment.** We use four real-world datasets: the climate dataset USHCN and three physiological datasets: PhysioNet2012, MIMIC-III, and MIMIC-IV. Following previous works (Yalavarthi

Table 1: Comparing njNLL values for probabilistic forecasting of irregularly sampled time series. Lower the better, best results in bold, second best in italics. For all four datasets `moses` is the best guaranteed consistent model.

| | Model | USHCN | PhysioNet2012 | MIMIC-III | MIMIC-IV |
|---|---|---|---|---|---|
| inconsistent | ProFITi | *-3.226 ± 0.225* | **-0.647 ± 0.078** | **-0.377 ± 0.032** | **-1.777 ± 0.066** |
| | Tactis-2 | -0.600 ± 0.082 | 0.017 ± 0.001 | 0.005 ± 0.001 | -0.012 ± 0.001 |
| consistent univariate | GRU-ODE | 0.766 ± 0.159 | 0.501 ± 0.001 | 0.961 ± 0.064 | 0.823 ± 0.318 |
| | NeuralFlows | 0.775 ± 0.152 | 0.496 ± 0.001 | 0.998 ± 0.177 | 0.689 ± 0.087 |
| | CRU | 0.761 ± 0.191 | 1.057 ± 0.007 | 1.234 ± 0.076 | OOM |
| | Tripletformer+ | 4.632 ± 8.179 | 0.519 ± 0.112 | 1.051 ± 0.141 | 0.686 ± 0.115 |
| consistent multivariate | GPR | 2.011 ± 1.376 | 1.367 ± 0.074 | 3.146 ± 0.359 | 2.789 ± 0.057 |
| | GMM | 1.050 ± 0.031 | 1.063 ± 0.002 | 1.160 ± 0.020 | 1.076 ± 0.003 |
| | `moses` (ours) | **-3.357 ± 0.176** | *-0.491 ± 0.041* | *-0.305 ± 0.027* | *-1.668 ± 0.097* |

et al., 2024b; Biloš et al., 2021), we observe the first 36h and predict the next 3 time steps for the physiological datasets, and observe 3 years and forecast the next 3 time steps for the USHCN dataset. Basic statistics of the datasets are provided in Appendix B. Although we predict the next 3 time steps, the number of observations $N$ and queries $K$ vary (see Table 3). We split the data into Train, Validation, and Test sets in a $70 : 10 : 20$ ratio. We trained `moses` using Adam optimizer, with learning rate 0.001 and batch size of 64. We search for hyperparameters, including the number of mixture components $D \in \{1, 2, 5, 7, 10\}$, attention heads $\in \{1, 2, 4\}$, and latent embedding $M, F \in \{16, 32, 64, 128\}$. All models are implemented in PyTorch and run on GeForce RTX 3090 and GTX 1080 Ti GPUs.

**Baselines.** As baseline models, we use NeuralFlows (Biloš et al., 2021), GRU-ODE (De Brouwer et al., 2019), CRU (Schirmer et al., 2022), GPR (Dürichen et al., 2015), ProFITi (Yalavarthi et al., 2024b) and TACTiS-2 (Ashok et al., 2024). Our encoder is similar to Tripletformer, predicting marginal distributions. Hence, we predict the mean and variance of a Gaussian distribution for $\widetilde{h}_k$, calling the model Tripletformer+. NeuralFlows, GRU-ODE, CRU, and Tripletformer+ predict only marginals and are marginalization consistent, as their joint distribution is the product of marginals. GPR is also marginalization consistent. We also compare with Gaussian Mixture Model (GMM) which is the `moses` without the flows attached to highlight the advantage of flows in `moses`.

**Evaluation metric for multivariate distribution.** We use normalized joint negative log-likelihood (njNLL) (Yalavarthi et al., 2024b) as the evaluation metric (15). njNLL generalizes the joint negative log-likelihood of varying number of variables. We note that CRPS, widely used metric for probabilistic univariate forecasting, cannot be applied for multivariate distributions. Also, sampling based metrics for multivariate distributions like Energy Score not only suffers from the curse of dimensionality but also cannot evaluate the forecasts properly (Marcotte et al., 2023). On the other hand, another metric CRPS-sum (Rasul et al., 2021) was shown to provide misleading indication of the performance (Koochali et al., 2022). In their study CRPS-sum cannot distinguish the predictions of a state-of-the-art model from random noise. Hence, we evaluate using njNLL in our experiments.

**Results for predicting joint distribution.** We compare `moses` with the published results from (Yalavarthi et al., 2024b) in Table 1. `moses` performs better than all the marginalization consistent models and comparable to non-marginalization consistent model ProFITi. Figure 5 shows njNLL vs marginal inconsistency (MI). `moses` not only achieves similar likelihoods as ProFITi, its MI is close to 0 where ProFITi is up to an order of magnitude larger. Smaller values of MI for `moses` is due to sampling. We rounded the smaller MI to 0.1. TACTiS, despite not providing consistency guarantees, turns out to be the most consistent model among those without such guarantees. We hypothesize that TACTiS may just learn a (for each query size) constant copula that actually does not depend on the queries; we leave this analysis to future research on copula models.

**Comparing for marginals: demonstrating the importance of marginalization consistency.** We evaluate the models using marginal Negative Log-Likelihood (mNLL), a common metric in prior

Table 2: Trained for njNLL and evaluate for Marginal Negative Log-likelihood (mNLL), lower the better. Demonstrates the advantage of marginalization consistency.

|  | USHCN | PhysioNet'12 | MIMIC-III | MIMIC-IV |
|---|---|---|---|---|
| GRU-ODE | $0.776 \pm 0.172$ | $0.504 \pm 0.061$ | $0.839 \pm 0.030$ | $0.876 \pm 0.589$ |
| Neural-Flows | $0.775 \pm 0.180$ | $0.492 \pm 0.029$ | $0.866 \pm 0.097$ | $0.796 \pm 0.053$ |
| CRU | $0.762 \pm 0.180$ | $0.931 \pm 0.019$ | $1.209 \pm 0.044$ | OOM |
| ProFITi-marg | *-3.324 ± 0.206* | *-0.016 ± 0.085* | *0.408 ± 0.030* | *0.500 ± 0.322* |
| moses (ours) | **-3.355 ± 0.156** | **-0.271 ± 0.028** | **0.163 ± 0.026** | **-0.634 ± 0.017** |

work (Biloš et al., 2021; Schirmer et al., 2022). Given test data $\mathcal{D}^{\text{TEST}}$, mNLL is the average NLL of future values (marginals). Table 2 shows results for the top models in each category from Table 1. ProFITi-marg refers to ProFITi querying one variable ($k = 1$) in a single pass, requiring $K = 5$ runs for a 5-variable series. Notably, moses outperforms all baselines, including ProFITi-marg, in mNLL, despite ProFITi offering superior njNLL, due to the marginalization inconsistency of ProFITi.

**Computational Complexity.** The scalability of our model is determined by multiple factors: the number of mixture components $D$, the number of query points $K$, and the dimensionality of the latent space $M$. By design, the normalizing flows are separable and therefore have diagonal Jacobians, hence can be computed in $\mathcal{O}(MK)$. The main bottleneck of the method is computing the density of the multivariate base distribution, which requires computing the determinant of the $K \times K$ matrices $\Sigma_d$, as well as a quadratic term $y^T \Sigma_d^{-1} y$. Generally, this costs $\mathcal{O}(K^3)$, however, thanks to the low rank representation $\Sigma_d = \mathbb{I}_K + U_d U_d^T$ in (13b), it can be done in $\mathcal{O}(M^2 K)$, as shown in Appendix A.5. Hence, the overall model complexity is $\mathcal{O}(M^2 KD)$, i.e. linear in the query length $K$.

## CONCLUSIONS

In this work, we propose, moses: marginalization consistent mixture of separable flows, a novel model for probabilistic forecasting of irregular time series. We showed how to parametrize various components of moses such that it is decomposable and marginalization consistent. Our experimental results on 4 real-world irregularly sampled time series datasets show that moses not only performs similar to state-of-the-art ProFITi model but unlike ProFITi moses is marginalization consistent.

## REPRODUCIBILITY STATEMENT

To ensure reproducibility, an implementation of moses in PyTorch is publicly available at https://anonymous.4open.science/r/seperable_flows-BACC. Detailed proofs for all lemmas introduced in the paper can be found in the Appendix.

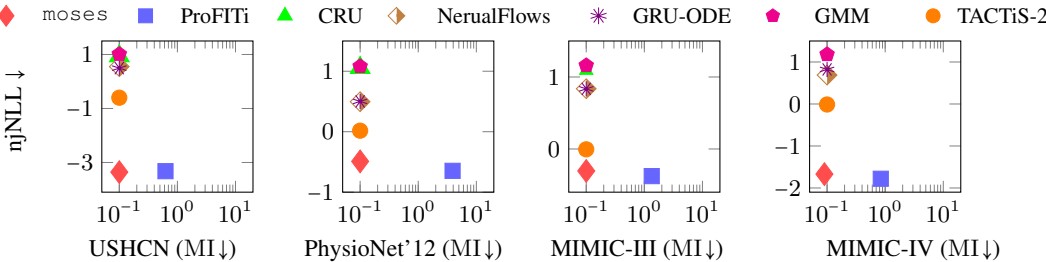

Figure 5: njNLL vs. MI. The Marginalization Inconsistency (17) compares predicted marginal distributions of individual queries to the numerically integrated joint distribution. moses is marginalization consistent within sampling error, where small values arise from sampling.

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

# A  THEORY

## A.1  PROOF OF LEMMA 1

*Proof.* Since $X$ is a common conditional to all the marginals, we can ignore it. So, assume that $f$ is a separable transformation:

$$f(z \mid Q) = (\phi(z_1 \mid Q_1), \dots, \phi(z_K \mid Q_K)) \tag{18}$$

and that $\hat{p}_Z(z \mid Q)$ is marginalization consistent model. Then, the predictive distribution is

$$\hat{p}(y \mid Q) = \hat{p}_Z(f^{-1}(y \mid Q) \mid Q) \cdot \left| \det \frac{\mathrm{d}f^{-1}(y \mid Q)}{\mathrm{d}y} \right| \tag{19}$$

Since $f$ is separable, it follows that the Jacobian is diagonal:

$$\frac{\mathrm{d} f^{-1}(y \mid Q)}{\mathrm{d} y} = \frac{\mathrm{d}\left(\phi^{-1}(y_1 \mid Q_1), \ldots \phi^{-1}(y_1 \mid Q_1)\right)}{\mathrm{d}(y_1, \ldots, y_K)} \tag{20}$$

$$= \mathrm{diag}\left(\frac{\mathrm{d}\phi^{-1}(y_1 \mid Q_1)}{\mathrm{d} y_1}, \ldots, \frac{\mathrm{d}\phi^{-1}(y_K \mid Q_K)}{\mathrm{d} y_K}\right) \tag{21}$$

Hence, the determinant of the Jacobian is the product of the diagonal elements:

$$\left|\det \frac{\mathrm{d} f^{-1}(y \mid Q)}{\mathrm{d} y}\right| = \prod_{k=1:K}\left|\det \frac{\mathrm{d}\phi^{-1}(y_k \mid Q_k)}{\mathrm{d} y_k}\right| = \prod_{k=1:K}\left|\frac{\mathrm{d}\phi^{-1}(y_k \mid Q_k)}{\mathrm{d} y_k}\right| \tag{22}$$

Using this fact, we can integrate the joint density over $y_k$ to get the marginal density:

$$\int \hat{p}(y \mid Q)\,\mathrm{d} y_k = \int \hat{p}_Z(f^{-1}(y \mid Q) \mid Q) \cdot \left|\det \frac{\mathrm{d} f^{-1}(y \mid Q)}{\mathrm{d} y}\right|\mathrm{d} y_k \qquad \rhd \; (7)$$

$$= \int \hat{p}_Z(f^{-1}(y \mid Q) \mid Q) \cdot \prod_{k=1:K}\left|\frac{\mathrm{d}\phi^{-1}(y_k \mid Q_k)}{\mathrm{d} y_k}\right|\mathrm{d} y_k \qquad \rhd \; (22)$$

$$= \left(\prod_{l \neq k}\left|\frac{\mathrm{d}\phi^{-1}(y_k \mid Q_k)}{\mathrm{d} y_k}\right|\right) \cdot \int \hat{p}_Z(f^{-1}(y \mid Q) \mid Q) \cdot \left|\frac{\mathrm{d}\phi^{-1}(y_k \mid Q_k)}{\mathrm{d} y_k}\right|\mathrm{d} y_k$$

$$= \left(\prod_{l \neq k}\left|\frac{\mathrm{d}\phi^{-1}(y_k \mid Q_k)}{\mathrm{d} y_k}\right|\right) \cdot \int \hat{p}_Z(z \mid Q)\,\mathrm{d} z_k \qquad \rhd \; \text{transf.-thm}$$

$$= \left(\prod_{l \neq k}\left|\frac{\mathrm{d}\phi^{-1}(y_k \mid Q_k)}{\mathrm{d} y_k}\right|\right) \hat{p}_Z(z_{-k} \mid Q_{-k}) \qquad \rhd \; (5)$$

$$= \hat{p}_Z(z_{-k} \mid Q_{-k})\left|\det \frac{\mathrm{d} f^{-1}(y_{-k} \mid Q_{-k})}{\mathrm{d} y_{-k}}\right| \qquad \rhd \; (22)$$

$$= \hat{p}(y_{-k} \mid Q_{-k}) \qquad \rhd \; (7)$$

□

## A.2 PROOF OF LEMMA 2

*Proof.* Consider a mixture model of the form

$$\hat{p}(y \mid Q, X) := \sum_{d=1}^{D} w_d(X)\hat{p}_d(y \mid Q, X) \tag{23}$$

satisfying the conditions from Lemma 2, i.e. the component models $\hat{p}_d$ satisfy the requirements **R1**-**R3** and the weight function $w: \mathrm{Seq}(\mathcal{X}) \to \Delta^D$ is permutation invariant with respect to $X$.

1. $\hat{p}$ satisfies **R1**: By construction of the mixture model, it has the same domain and codomain as the component models.

2. $\hat{p}$ satisfies **R2**: Let $\pi \in S_{|Q|}$ and $\tau \in S_{|X|}$, then

$$\hat{p}(y \mid Q^\pi, X^\tau) = \sum_{d=1}^{D} w_d(X^\tau)\hat{p}_d(y \mid Q^\pi, X^\tau)$$

$$= \sum_{d=1}^{D} w_d(X)\hat{p}_d(y \mid Q, X) \qquad \rhd \; \text{permutation invariance of } w \text{ and } \hat{p}_d$$

$$= \hat{p}(y \mid Q, X)$$

3. $\hat{p}$ satisfies **R3**:

$$\int \hat{p}(y \mid Q, X) \, \mathrm{d}y_k = \int \sum_{d=1}^{D} w_d(X) \hat{p}_d(y \mid Q, X) \, \mathrm{d}y_k$$

$$= \sum_{d=1}^{D} w_d(X) \int p_d(y \mid Q, X) \, \mathrm{d}y_k$$

$$= \sum_{d=1}^{D} w_d(X) p_d(y_{-k} \mid Q_{-k}, X) \qquad \triangleright \hat{p}_d \text{ is marginalization consistent}$$

$$= p_Y(y_{-k} \mid Q_{-k}, X)$$

$\square$

### A.3 PROOF OF THEOREM 2

*Proof.* Due to Lemma 1, it is sufficient to show that all the component models satisfy the requirements **R1**-**R3**. Since we use Gaussian Processes as the base distribution, Lemma 1 ensures that each component model is marginalization consistent, establishing **R3**. Requirement **R1** is by construction. Finally, permutation invariance **R2** can be seen as follows:

First, note that, by Equation (12), it follows that if $\mathbf{h}^{\text{OBS}}$ is permutation equivariant with respect to $X$, and $\widetilde{\mathbf{h}}$ and $\mathbf{h}$ are both permutation equivariant with respect to $Q$ and permutation invariant with respect to $X$. Now, let $\pi \in S_{|Q|}$ and $\tau \in S_{|X|}$, then, for the $d$-th component model $\hat{p}_{Y_d}(y \mid Q, X)$. In particular, the flow satisfies $f_d^{-1}(y^\pi, Q^\pi, X^\tau) = f^{-1}(y^\pi, \mathbf{h}_d^\pi) = z^\pi$. Therefore:

$$\hat{p}_{Y_d}(y^\pi \mid Q^\pi, X^\tau) = \hat{p}_{Z_d}(f^{-1}(y^\pi, Q^\pi, X^\tau) \mid Q^\pi, X^\tau) \cdot \left| \det \frac{\mathrm{d}f^{-1}(y^\pi, Q^\pi, X^\tau)}{\mathrm{d}y^\pi} \right|$$

$$= \mathcal{N}(f^{-1}(y^\pi, \mathbf{h}_d^\pi) \mid \mu(\mathbf{h}_d^\pi), \Sigma(\mathbf{h}_d^\pi)) \cdot \left| \det \frac{\mathrm{d}f^{-1}(y^\pi, \mathbf{h}^\pi)}{\mathrm{d}y^\pi} \right| \quad \triangleright \text{ by remark above}$$

$$= \mathcal{N}(z^\pi \mid \mu(\mathbf{h}_d^\pi), \Sigma(\mathbf{h}_d^\pi)) \cdot \left| \det \frac{\mathrm{d}f^{-1}(y^\pi, \mathbf{h}^\pi)}{\mathrm{d}y^\pi} \right|$$

$$= \mathcal{N}(z \mid \mu, \Sigma) \cdot \left| \det \frac{\mathrm{d}f^{-1}(y^\pi, \mathbf{h}^\pi)}{\mathrm{d}y^\pi} \right| \quad \triangleright \text{ by permutation invariance of GP}$$

$$= \mathcal{N}(z \mid \mu, \Sigma) \cdot \left| \det \frac{\mathrm{d}f^{-1}(y, \mathbf{h})}{\mathrm{d}y} \right| \quad \triangleright \text{ by (22)}$$

$$= \hat{p}_{Y_d}(y \mid Q, X)$$

$\square$

### A.4 LINEAR RATIONAL SPLINES

Linear Rational Splines (LRS) are computationally efficient spline functions Dolatabadi et al. (2020). Formally, given a set of monotonically increasing points $\{(u_m, v_m)\}_{m=1:M}$ called knots, that is $u_m < u_{m+1}$ and $v_m < v_{m+1}$, along with their corresponding derivatives $\{\Delta_m > 0\}_{m=1:M}$, then the LRS transformation $\phi(u)$ within a bin $u \in [u_m, u_{m+1}]$ is:

$$\phi(u) = \begin{cases} \frac{\alpha_m v_m (\lambda_m - \tilde{u}) + \bar{\alpha}_m \bar{v}_m \tilde{u}}{\alpha_m (\lambda_m - \tilde{u}) + \bar{\alpha}_m \tilde{u}} & : \quad 0 \le \tilde{u} \le \lambda_m \\ \frac{\bar{\alpha}_m \bar{v}_m (1 - \tilde{u}) + \alpha_{m+1} v_{m+1} (\tilde{u} - \lambda_m)}{\bar{\alpha}_m (1 - \tilde{u}) + \alpha_{m+1} (\tilde{u} - \lambda_m)} & : \quad \lambda_m \le \tilde{u} \le 1 \end{cases} \quad \text{where} \quad \tilde{u} = \frac{u - u_m}{u_{m+1} - u_m} \in [0, 1] \quad (24)$$

Here, $\lambda_m \in (0, 1)$ signifies the location of automatically inserted virtual knot between $u_m$ and $u_{m+1}$ with value $\bar{v}_m$. The values of $\lambda_m, \alpha_m, \bar{\alpha}_m$ and $\bar{v}_m$ are all automatically derived from the original knots and their derivatives Dolatabadi et al. (2020). For a conditional LRS $\phi(z_k; \mathbf{h}_{d,k}, \theta)$, the function parameters such as width and height of each bin, the derivatives at the knots, and $\lambda$ are computed from the conditioning input $\mathbf{h}_{d,k}$ and some model parameters $\theta$. $\theta$ helps to project $\mathbf{h}_{d,k}$ to the function

Table 3: Statistics of the datasets used in our experiments. Sparsity means the percentage of missing observations in the time series. $N$ is the total number of observations and $K$ is the number of queries in our experiments in Section 7.

| Name | #Samples | #Channels | Sparsity | N | K |
|------|----------|-----------|----------|---|---|
| USHCN | 1100 | 5 | 77.9% | $8-322$ | $3-6$ |
| PhysioNet'12 | 12,000 | 37 | 85.7% | $3-519$ | $1-53$ |
| MIMIC-III | 21,000 | 96 | 94.2% | $4-709$ | $1-85$ |
| MIMIC-IV | 18,000 | 102 | 97.8% | $1-1382$ | $1-79$ |

parameters, and is common to all the variables $z_{1:K}$ so that the transformation $\phi$ can be applied for varying number of variables $K$. Additionally, we set $\theta$ common to all the components as well. Since, each component has separate embedding for a variable $z_k$ ($\mathbf{h}_{d,k}$), we achieve different transformations in different components for same variable.

In summary, the conditional flow model is separable across the query size $f = f_1 \times \cdots \times f_K$ with

$$f_d(y) := f(y \mid \mathbf{h}_d) = (\phi(y_1, \mathbf{h}_{d,1}), \ldots, \phi(y_K, \mathbf{h}_{d,K})) \tag{25}$$

### A.5 EFFICIENT COMPUTATION OF GAUSSIAN WITH LOW RANK PERTURBATION

**Lemma 3.** Given a multivariate Normal distribution $\mathcal{N}(\mu, \Sigma)$, where $\Sigma = \mathbb{I}_K + UU^T$ with $U \in \mathbb{R}^{K \times M}$, and $K \geq M$, the density of the distribution can be computed in $\mathcal{O}(M^2 K)$ time.

*Proof.* The density of the distribution is given by:

$$p(z) = \frac{1}{(2\pi)^{K/2} |\Sigma|^{1/2}} \exp\left(-\frac{1}{2}(z-\mu)^T \Sigma^{-1}(z-\mu)\right)$$

By the matrix determinant lemma, also known as Weinstein–Aronszajn identity, we have $\det(\mathbb{I}_K + UU^T) = \det(\mathbb{I}_M + U^T U)$, and since the determinant of a $n \times n$ matrix can be computed in $\mathcal{O}(n^3)$ time, we can compute the determinant of $\Sigma$ in $\mathcal{O}(M^3) \leq \mathcal{O}(M^2 K)$ time.

Moreover, the quadratic term $y^T \Sigma^{-1} y$ can also be computed in $\mathcal{O}(M^2 K)$ as follows: First, by the matrix inversion lemma, also known as Woodbury identity, $(\mathbb{I}_K + UU^T)^{-1} = \mathbb{I}_K - U(\mathbb{I}_M + U^T U)^{-1} U^T$. Hence

$$y^T \Sigma^{-1} y = y^T (\mathbb{I}_K - U(\mathbb{I}_M + U^T U)^{-1} U^T) y = \|y\|^2 - v^T (\mathbb{I}_M + U^T U)^{-1} v$$

where $v = U^T y \in \mathbb{R}^M$ costs $\mathcal{O}(M \cdot K)$, the reduced matrix $\Sigma' = \mathbb{I}_M + U^T U$ can be created in $\mathcal{O}(M^2 K)$ and the quadratic term $v^T(\Sigma')^{-1} v$ can be computed in $\mathcal{O}(M^3)$. $\qed$

## B DATASETS

4 real-world datasets are used in the experiments.

**USHCN Menne et al. (2015).** This is a climate dataset consisting of 5 climate variables such as daily temperatures, precipitation and snow measured over 150 years at 1218 meteorological stations in the USA. Following De Brouwer et al. (2019); Yalavarthi et al. (2024b), we selected 1114 stations and an observation window of 4 years from 1996 until 2000.

**PhysioNet2012 Silva et al. (2012).** This physiological dataset consists of the medical records of 12,000 patients who are admitted into ICU. 37 vitals are recorded for 48 hrs. Following the protocol of Yalavarthi et al. (2024a); Che et al. (2018), dataset consists of hourly observations in each series.

**MIMIC-III Johnson et al. (2016).** This is also a physiological dataset. It is a collection of readings of the vitals of the patients admitted to ICU at Beth Israeli Hospital. Dataset consists of 18,000 instances and 96 variables are measured for 48 hours. Following De Brouwer et al. (2019); Biloš et al. (2021); Yalavarthi et al. (2024b) observations are rounded to 30 minute intervals.

Table 4: Comparing models w.r.t. CRPS score on marginals. Lower the better.

| Model | USHCN | PhysioNet'12 | MIMIC-III | MIMIC-IV |
|---|---|---|---|---|
| GRU-ODE | 0.313 ± 0.012 | 0.278 ± 0.001 | 0.308 ± 0.005 | 0.281 ± 0.004 |
| Neural-flows | 0.306 ± 0.028 | 0.277 ± 0.003 | 0.308 ± 0.004 | 0.281 ± 0.004 |
| CRU | 0.247 ± 0.010 | 0.363 ± 0.002 | 0.410 ± 0.005 | OOM |
| TACTiS-2 | 0.200 ± 0.023 | 0.363 ± 0.005 | 0.410 ± 0.009 | 0.366 ± 0.008 |
| ProFITi | **0.183 ± 0.009** | *0.268 ± 0.002* | **0.295 ± 0.002** | **0.226 ± 0.002** |
| ProFITi-marg | *0.182 ± 0.007* | 0.271 ± 0.003 | 0.319 ± 0.003 | 0.279 ± 0.012 |
| moses (ours) | 0.220 ± 0.019 | **0.260 ± 0.002** | 0.296 ± 0.005 | *0.245 ± 0.010* |

Table 5: Comparing models w.r.t. MSE. Lower the better.

| Model | USHCN | PhysioNet'12 | MIMIC-III | MIMIC-IV |
|---|---|---|---|---|
| GRU-ODE | 0.410 ± 0.106 | 0.329 ± 0.004 | 0.479 ± 0.044 | 0.365 ± 0.012 |
| Neural-Flows | 0.424 ± 0.110 | 0.331 ± 0.006 | 0.479 ± 0.045 | 0.374 ± 0.017 |
| CRU | *0.290 ± 0.060* | 0.475 ± 0.015 | 0.725 ± 0.037 | OOM |
| GraFITi | **0.256 ± 0.027** | **0.286 ± 0.001** | **0.401 ± 0.028** | **0.233 ± 0.005** |
| Tripletformer+ | 0.349 ± 0.131 | 0.293 ± 0.018 | 0.547 ± 0.068 | 0.369 ± 0.030 |
| TACTiS-2 | 0.381 ± 0.127 | 0.474 ± 0.006 | 0.759 ± 0.071 | 0.578 ± 0.034 |
| ProFITi | 0.321 ± 0.041 | *0.299 ± 0.007* | *0.495 ± 0.075* | *0.268 ± 0.007* |
| ProFITi-marg | 0.308 ± 0.061 | 0.305 ± 0.007 | 0.548 ± 0.063 | 0.389 ± 0.015 |
| moses (ours) | 0.411 ± 0.099 | 0.307 ± 0.006 | 0.517 ± 0.057 | 0.342 ± 0.028 |

**MIMIC-IV Johnson et al. (2021).** The successor of the MIMIC-III dataset. Here, 102 variables from patients admitted to ICU at a tertiary academic medical center in Boston are measured for 48 hours. Following De Brouwer et al. (2019); Biloš et al. (2021); Yalavarthi et al. (2024b) observations, are rounded to 1 minute intervals.

**Blast distribution (toy dataset).** Blast distribution is a bivariate distribution which is created as follows:

$$z \sim \mathcal{N}\left(\begin{bmatrix} 0 \\ 0 \end{bmatrix}, \begin{bmatrix} 1 & 1 \\ 1 & 2 \end{bmatrix}\right)$$
$$y = \text{sign}(z) \odot z \odot z$$

**Circle (toy dataset).** Circle is also a bi-variate distribution.

$$z \sim \mathcal{N}(0, \mathbb{I}_2)$$
$$y = \frac{z}{\|z\|_2} + 0.05 \cdot \mathcal{N}(0, \mathbb{I}_2)$$

## C  ADDITIONAL EXPERIMENTS

### C.1  COMPARING FOR MARGINALS IN TERMS OF CRPS

We compare with CRPS score in Table 4, a widely used evaluation metric in time series forecasting. We see that moses outperforms all the consistent models. It performs better than ProFITi-marg in 3 out of 4 dataset. For ProFITi, ProFITi-marg and moses, we sampled 1000 instances and computed the CRPS. Due to marginal inconsistency in ProFITi, the CRPS scores of ProFITi are vastly different from that of ProFITi-marg in PhysioNet'12, MIMIC-III and MIMIC-IV datasets.

### C.2  COMPARING FOR POINT FORECASTING

Here, we would like to see how the models compare with point forecasting. We use mean squared error as the evaluation metric. Here, we also compare with GraFITi Yalavarthi et al. (2024a), state-of-the-art point forecasting model for irregularly sampled time series. Results are presented in Table 5.

Table 6: Experiment on varying observation and forecast horizons. Evaluation metric-njNLL, Lower the better

|  | 36/12 | 24/24 | 12/36 |
|---|---|---|---|
| NeuralFlows | 0.709 ± 0.483 | 1.097 ± 0.044 | 1.436 ± 0.187 |
| ProFITi | **-0.768 ± 0.041** | **-0.355 ± 0.243** | **-0.291 ± 0.415** |
| moses | *-0.315 ± 0.016* | *-0.298 ± 0.027* | *-0.063 ± 0.049* |

Table 7: Comparing for Energy Score. Lower the better

|  | USHCN | PhysioNet'12 | MIMIC-III | MIMIC-IV |
|---|---|---|---|---|
| NeuralFlows | 0.661 ± 0.059 | 1.691 ± 0.001 | 1.381 ± 0.033 | 0.982 ± 0.009 |
| ProFITi | 0.452 ± 0.044 | 0.879 ± 0.303 | 1.606 ± 0.168 | 0.808 ± 0.003 |
| moses | 0.552 ± 0.044 | 1.599 ± 0.013 | 1.353 ± 0.033 | 0.906 ± 0.029 |

While GraFITi continues to be the best, ProFITi is the second best. moses has similar performance as ProFITi in PhysioNet'12 and MIMIC-III datasets. Both GraFITi and ProFITi are inconsistent models.

Interesting to observe that moses is significantly better than Tripletformer+ in MIMIC-III and MIMIC-IV datasets and have similar results in USHCN and PhysioNet'12 datasets. Also, it is better than existing consistent models GRU-ODE De Brouwer et al. (2019), Neural Flows Biloš et al. (2021) and CRU Schirmer et al. (2022). Note that, it is often observed in the literature that models for uncertainty quantification quite often suffer from somewhat worse point forecasts (Lakshminarayanan et al., 2017; Seitzer et al., 2021).

## C.3 Experiment on varying observation and forecast horizons

We would like to see if moses is scalable to long observationa and forecast horizons. For, we performed an experiment on varying length observation and forecasting horizons on Physionet'12 dataset and compared against the published results from (Yalavarthi et al., 2024b) in Table 6. The observation and forecasting horizons are: {(36h, 12h), (24h, 36h), (12h, 26h)}. We see that moses performs better than best consistent model Neural Flows for all the observation and forecasting horizons. However, ProFITi is the best performing model.

We stress that ensuring that a model is marginalization-consistent and allows for tractable marginals imposes significant restrictions on the modeling, which may lead to a slight decrease in performance when predicting the joint distribution.

## C.4 Comparing for Energy Score

The Energy Score between the ground truth $y$ and predicted distribution $\hat{p}_Y$ is computed as:

$$\text{ES}(y, \hat{p}_Y) := \mathop{\mathbb{E}}_{y' \sim \hat{p}_Y} \|y - y'\|_2^p - \frac{1}{2} \mathop{\mathbb{E}}_{y', y'' \sim \hat{p}_Y} \|y' - y''\|_2^p, \tag{26}$$

where $\|\cdot\|_2$ denotes the Euclidean norm and $p \in (0, 2)$ is a parameter. In our evaluation, we set $p = 1$. Marcotte et al. (2023) demonstrated that the Energy Score is not a reliable metric for evaluating multivariate distributions. Additionally, it suffers from the curse of dimensionality, as it requires $N^K$ samples, where $K$ is the number of variables and $N$ is the number of samples required to accurately estimate a univariate distribution.

However, since many regularly sampled, fully observed multivariate time series probabilistic forecasting models use the Energy Score as an evaluation metric, we examine how moses compares to the best-performing inconsistent multivariate probabilistic model, ProFITi, and the consistent univariate probabilistic model, NeuralFlows in Table 7. Our results show that moses outperforms NeuralFlows across all datasets. As shown by the njNLL metric in Table 1, ProFITi is the best-performing model, outperforming moses in 3 out of 4 datasets.

Table 9: Comparing NLL values for density estimation of tabular data. Lower the better, best results in bold, second best in italics.

| Model | power | gas | miniboone | hepmass |
|---|---|---|---|---|
| MADE Germain et al. (2015) | 3.08 ± 0.03 | -3.56 ± 0.04 | 15.59 ± 0.50 | 20.98 ± 0.02 |
| Real NVP Dinh et al. (2017) | 0.02 ± 0.01 | -4.78 ± 1.80 | *13.55 ± 0.49* | *19.62 ± 0.02* |
| MAF Papamakarios et al. (2017) | -0.14 ± 0.01 | **-9.07 ± 0.02** | **11.75 ± 0.44** | **17.70 ± 0.02** |
| EinsumNet Sidheekh et al. (2023) | *-0.20 ± 0.01* | -3.57 ± 0.08 | 35.93 ± 0.06 | 22.79 ± 0.05 |
| Einsum+LRS Sidheekh et al. (2023) | **-0.36 ± 0.01** | -4.79 ± 0.04 | 34.21 ± 0.01 | 22.46 ± 0.01 |
| `moses` | -0.10 ± 0.01 | *-5.87 ± 0.27* | 22.66 ± 2.08 | 23.95 ± 0.20 |

## C.5 ABLATION STUDY.

Table 8: Ablation study on PhysioNet2012

| Model | njNLL ($\downarrow$) |
|---|---|
| `moses` | -0.491 ± 0.041 |
| `moses`$-f$ | 1.063 ± 0.002 |
| `moses`$-$COV | -0.308 ± 0.024 |
| `moses`$-w$ | -0.451 ± 0.038 |
| `moses` (1) | -0.493 ± 0.029 |

Using PhysioNet2012, we show the importance of different model components. As summarized in Table 8, the performance is reduced by removing the flows ($\texttt{moses} - f$) which is same as GMM. It is expected that normalizing flows are more expressive compared to simple mixture of Gaussians. On the other hand, by using only isotropic Gaussian as the base distribution ($\texttt{moses} - $COV) model performance worsened. Similarly, parameterizing the components weights have a slight advantage over fixing them to $1/D$ with $D$ being the number of components. One interesting observation is even using single component ($\texttt{moses}(1)$) gives similar results compared to mixture of such components. This could be because the dataset we have may not require multiple components. We note that we have $D = 1$ in our hyperparameter space, and we select the best $D$ based on validation dataset.

## C.6 RESULTS FOR UNCONDITIONAL DENSITY ESTIMATION ON TABULAR DATA

A family of models known as *Probabilistic Circuits* (Choi et al., 2020) has been shown to support both marginal tractability and marginalization consistency, making them suitable for density estimation with a fixed number of variables. Therefore, comparing `moses` against these models is interesting for evaluating its effectiveness on similar tasks.

Specifically, we compare `moses` with two circuit-based models—EinsumNet (Choi et al., 2020) and EinsumNet+LRS (Sidheekh et al., 2023)—as well as with other well-known density estimation models. Unlike the above models, `moses` is not restricted to a fixed number of variables.

The comparison uses published results from (Sidheekh et al., 2023) across four tabular datasets: **power**, **gas**, **miniboone**, and **hepmass**. We follow the same data-splitting protocol used in previous work (Papamakarios et al., 2017; Sidheekh et al., 2023). Since this is not conditional density estimation, trainable parameters include univariate transformations, mixture weights $w$, and the base distribution parameters $(\mu, \Sigma)$.

Table 9 shows the Negative Log-Likelihood (NLL) results for all models. While MAF achieves the best overall performance, `moses` outperforms Real NVP on 2 out of the 4 datasets and exceeds the best decomposable circuit model, EinsumNet+LRS, on another 2 datasets. This demonstrates `moses`'s competitive performance, especially given that it is not restricted to a fixed variable count, unlike the other models.

## D  SIMPLEQUATIONS

$$\text{log-likelihood-term} = (\alpha - 1)\log(y) + (\beta - 1)\log(1 - y)$$
$$\text{norm factor} = \log(\gamma(\alpha)) + \log(\gamma(\beta)) - \log(\gamma(\alpha + \beta))$$
$$\text{loss} = \text{log-likelihood-term} - \text{norm factor}$$

$$\text{log-likelihood-term} = -\frac{(\text{logit}(y) - \mu)^2}{2\sigma^2}$$
$$\text{norm factor} = 0.5\left(\log(2\pi) + \log(\sigma^2)\right) + \log(y) + \log(1 - y)$$
$$\text{loss} = \text{log-likelihood-term} - \text{norm factor}$$

$$\text{weighted log-likelihood} = \frac{\sum_{kw=1}^{N} \text{clicks}(kw) \cdot \log \hat{p}_{\text{CNR}}(cnr \mid kw)}{\sum_{kw=1}^{N} \text{clicks}(kw)} \tag{27}$$

$$\text{log-likelihood-term} = -\frac{(\log(y) - \mu)^2}{2\sigma^2}$$
$$\text{norm factor} = 0.5\left(\log(2\pi) + \log(\sigma^2)\right) + \log(y)$$
$$\text{loss} = \text{log-likelihood-term} - \text{norm factor}$$

$$\text{log-likelihood-term} = -\frac{(y - \mu)^2}{2\sigma^2}$$
$$\text{norm factor} = 0.5\left(\log(2\pi) + \log(\sigma^2)\right)$$
$$\text{loss} = \text{log-likelihood-term} - \text{norm factor}$$

$$\frac{\sum_{kw=1}^{N} \text{clicks}(kw) \cdot (cnr \log(\hat{cnr}) + (1 - cnr)\log(1 - \hat{cnr}))}{\sum_{kw=1}^{N} \text{clicks}(kw)} \tag{28}$$

$$\sqrt{\frac{\sum_{kw=1}^{N} \text{clicks}(kw) \cdot (cnr - \hat{cnr})^2}{\sum_{kw=1}^{N} \text{clicks}(kw)}} \tag{29}$$

