# OpenReview forum: "Marginalization Consistent Mixture of Separable Flows for Probabilistic Irregular Time Series Forecasting"
_ICLR.cc/2025/Conference — ICLR 2025 Conference Withdrawn Submission_

### Official Review · Reviewer_9wkz · 2024-11-02

**Soundness:** 2
**Presentation:** 2
**Contribution:** 2
**Rating:** 5
**Confidence:** 2

**Summary:**

This paper proposes a new model, called moses (marginalization consistent mixture of separable flows) for probabilistic forecasting of irregular time series. In particular, the proposed model mixes several normalizing flows with Gaussian Processes with full covariance matrix as source distributions and a separable invertible transformation. Some empirical results are provided to demonstrate the effectiveness of moses in achieving marginalization consistency and lower Marginal Negative Log-likelihood when compared to existing methods.

**Strengths:**

- Overall the paper is well written and the presentation is easy to follow
- The design of the proposed marginalization consistent model that combines simple, separable transforms with a richer source distribution (a Gaussian Process with full covariance matrix) seems innovative

**Weaknesses:**

- Empirical results are limited to simple tasks and it is not clear how effective the proposed method would work on longer temporal sequences with higher spatial resolution
- The importance of marginalization consistency is not adequately demonstrated across different task settings. In other words, the paper does not address and perform detailed empirical studies to investigate how effective is maintaining marginalization consistency in improving the considered metrics for different lengths of training sequences, different lengths of forecast horizons and different dimensions (spatial resolutions) of training sequences
- Missing details on the training of the model: choices of optimization algorithm, batch size, learning rate, choices of architectural hyperparameters, etc. for each of the considered task

**Questions:**

- To empirically demonstrate marginalization consistency, instead of using the proposed two simple bivariate distributions, can we use simulated data from nonlinear (possibly chaotic) dynamical systems (with possibly complex stationary distributions)? I believe testing the proposed method on various such datasets could help to strengthen the paper
- How effectively would the proposed method perform for longer-horizon forecasts? Will the maintenance of marginalization consistency help improving longer term forecasting?

---

> ### Author Response · Authors · 2024-11-25
> **Response to reviewer 9wkz**
>
> Thank you for the insightful review.
>
> ### W1. Experiments on varying forecasting, observation and spatial range.
> We conducted experiments by varying the observation and forecasting times (obs/forc), and we have added these details to the supplementary sec. C.4. Our experiments cover tabular time series with up to 102 channels; spatio-temporal problem are outside the scope of this work.
>
> | model           | 36h/12h          | 24h/24h          | 12h/36h          |
> |-----------------|------------------|------------------|------------------|
> | **NeuralFlows** | 0.709±0.483      | 1.097±0.044      | 1.436±0.187      |
> | **ProFITi**     | **-0.768±0.041** | **-0.355±0.243** | **-0.291±0.415** |
> | **moses**       | *-0.315±0.016*   | *-0.298±0.027*   | *-0.063±0.049*   |
>
> Here, the evaluation metric is njNLL (lower the better). Similar to results in table 1, ProFITi is the best performing model for predicting joint NLL whereas moses is the second best.
>
> ### W2. Importance of marginalization consistency
> We showed the importance of marginalization consistency with the results in Tables 1 and 2. While our model, moses, has slightly worse results compared to ProFITi in predicting joint distributions — $\sim$5% worse likelihoods (note: in terms of actual density, not NLL; see table 1) — moses’ predictions for univariate marginal distributions are significantly better than ProFITi (~51%, see table 2) which is a consequence of ProFITi not adhering to marginalization consistency (fig. 4,5).
>
> Table 2 demonstrates that moses, trained for the joint NLL, makes very good predictions for the marginal variables, despite only being trained for the joint NLL, which is a direct consequence of the marginalization consistency.
> ### W3. Missing details on the training of the model
> Our model is trained using the Adam optimizer, with a batch size of 64 and a learning rate of 0.001. We added these details to the Experiments section. Other hyperparameters used in the model have already been provided in the paper.
> ### Q1. Demonstration of Marginalization consistency on synthetic ODE dataset
> Marginalization consistency does not have to be verified for specific datasets, as it is guaranteed by design. While we agree that the chaotic ODE dataset is an interesting dataset for comparisons, we believe that testing marginalization consistency on an ODE dataset does not add significant value beyond the bivariate case. Visualizing joint distributions is feasible in bivariate cases but becomes much more challenging in higher dimensions. Therefore, we do not plan to pursue this suggestion further.
> ### Q2. Forecasting on longer forecasting horizon, possibility of improving forecasting results with marginalization consistency
> As noted in response to W1, we added an experiment that varies the observation and forecasting horizons and compared our model’s results with the published results of ProFITi.
>
> Regarding the effect of marginalization consistency on forecasting accuracy, we would like to emphasize that maintaining marginalization consistency imposes certain constraints on the model. This results in a slight decrease in performance of joint NLL compared to models without this consistency, such as ProFITi, when compared on certain queries without their sub-queries.  For metrics that penalize violations of marginalization consistency (i.e. aggregating likelihoods over all sub-queries.) we expect our model to be better as seen in Table 2.

---

> > ### Comment · Reviewer_9wkz · 2024-11-26
> > **Thank you for the response**
> >
> > I thank the author(s) for the rebuttal. It seems like the proposed method (moses) could not outperform ProFITi on the experiments in W1. Also, it is still not clear to me what are the precise tradeoffs between imposing marginalization consistency and the forecasting performance. I am going to keep my score.

---

> > > ### Author Response · Authors · 2024-12-04
> > >
> > > Thanks for responding to our rebuttal. We address the questions you have:
> > > ### ProFITi vs moses performance comparison
> > > The primary utility of Moses lies not only in its ability to predict joint distributions but also in accurately predicting marginal distributions. As discussed in Section 1, we require a model capable of providing accurate marginal and joint predictions. While ProFITi excels at predicting the joint distributions it is trained on, its performance in predicting marginal distributions is notably poor. In contrast, Moses, though slightly less accurate than ProFITi in predicting joint distributions, significantly outperforms it in predicting marginal distributions, as demonstrated in Tables 1 and 2.
> > >
> > > We incorporated the results of ProFITi and Moses in predicting marginal distributions when trained on joint distributions, using varying observation and forecast horizons from the PhysioNet2012 dataset. The performance gap between joint and marginal distributions widened as the forecasting time increased. Moses's superior performance in predicting marginal distributions is largely attributed to its adherence to marginalization consistency.
> > >
> > > | model           | 36h/12h          | 24h/24h          | 12h/36h          |
> > > |-----------------|------------------|------------------|------------------|
> > > | **ProFITi-marg**     | 1.376±1.764 | 0.705±0.179 | 2.977±2.978|
> > > | **moses**       |  **-0.083±0.025**  | **-0.020±0.060**| **0.040±0.131** |
> > >
> > > ### Restrictions on a model for marginalization consistency
> > > 1. Encoder:
> > >
> > > To encode the conditioning input, such as observed time series and forecast queries, a probabilistic model that does not enforce marginalization consistency can use any suitable encoder. For example, ProFITi employs GraFITi, a state-of-the-art forecasting model for irregular time series. However, since GraFITi itself is not a consistent model—meaning that additional queries can produce different results for the same query—it cannot be directly applied to marginalization-consistent models like Moses.
> > >
> > > 2. Flows:
> > >
> > > To the best of our knowledge, univariate flows without channel interactions are crucial for preserving the decomposability and marginalization consistency of moses. This approach forms joint distributions by combining mixture components with base distributions with full covariance. In contrast, ProFITi leverages flows with channel interactions, enabling it to capture these relationships more effectively.
> > >
> > > As a deeper model, ProFITi excels at modeling channel interactions, while Moses is more constrained in this aspect, representing only "local" interactions through its mixture components.
> > >
> > > We will add this discussion to the revised version.
> > >
> > > Thank you once again for your thoughtful comments and engaging discussion. We hope we have addressed most of your questions and resolved any doubts you had.

---

### Official Review · Reviewer_Px4a · 2024-11-03

**Soundness:** 3
**Presentation:** 2
**Contribution:** 3
**Rating:** 6
**Confidence:** 3

**Summary:**

The work blends in the ideas of Gaussian Process Regression with Normalising Flows to develop a probabilistic forecasting model for joint distributions of targets in irregular time series. The proposed method shows the property of marginal consistency. The model can be queried in two ways; 1. by asking for joint predictive distribution of all variables and then marginalising out the variables not to be queried; 2. by directly querying the variable.

**Strengths:**

1. The paper focuses on a rather under-researched area providing a way for marginalisation consistency.
2. The standard deviation in the experimental results provides a good idea of the method's performance.
3. The idea of using mixtures of conditional normalising flows to counter expressivity issues is clever.

**Weaknesses:**

1. Could the author explain why we might want marginalisation consistency for models when we can directly query the desired variables and get the desired distribution?
2. Can authors comment on the performance of Moses if instead of mixtures of univariate normalising flows, a multivariate NF is used? This will surely violate the marginalisation consistency, but do we get better njNLL?
3. Is it possible to create prediction intervals on desired queries and attain the desired coverage due to marginalisation consistency?
4. Table 1 only compares njNLL, the metrics such as Energy Score were criticised, but it would be helpful to see a comparison of other such metrics.
5. The proposed methods seem to be lagging significantly wrt CRPS and MSE
6. A trivial question: I don't see a motivation for section C.3 in the appendix as one can expect Moses to not work that well for unconditional density estimation.

Edit: Thanks for the response, I have adjusted my score accordingly.

**Questions:**

See weaknesses

---

> ### Author Response · Authors · 2024-11-25
> **Response to reviewer Px4a**
>
> Thank you for your detailed review. Our responses are as follows:
> ### W1. Why do we need Marginalization Consistency?
> Marginalization consistency ensures that distributions over subsets of variables remain compatible with the full model, which enables coherent and reliable inference. Not satisfying marginalization consistency represents an internal inconsistency of the model. Would you trust a forecaster that makes self-contradictory predictions? Your question can also refer to downstream tasks for which consistent estimates of marginals and joint distributions would play a role: there are robust optimization or control problems for which this is true, e.g. [1].
> ### W2. Comparison of Moses using a multivariate Normalizing Flow?
> Replacing univariate flows with multivariate flows poses challenges due to the varying number of variables involved. To the best of our knowledge, only the ProFITi and Tactis models address this scenario, and serve as our baseline models.
> ### W3. Is it possible to create prediction intervals on desired queries and attain the desired coverage due to marginalisation consistency?
> The joint and marginal densities enable the calculation of prediction intervals for specific quantiles and allow for coverage testing. However, this process can become prohibitively expensive for large multivariate queries due to the curse of dimensionality.
> ### W4. Comparison w.r.t. Energy Score
> In the revised supplementary material, we provide a comparison among the best-performing models: Moses, ProFITi, and Neural Flows for Energy Score (lower the better). The results are as follows:
> |                 | USHCN           | PhysioNet'12    | MIMIC-III       | MIMIC-IV        |
> |-----------------|-----------------|-----------------|-----------------|-----------------|
> | **NeuralFlows** | 0.661±0.059     | 1.691±0.001     | 1.381±0.033     | 0.982±0.009     |
> | **ProFITi** | **0.452±0.044** | **0.879±0.303** | 1.606±0.168     | **0.808±0.003** |
> | **moses (our model)** | *0.552±0.044*     | *1.599±0.013*     | **1.353±0.033** | *0.906±0.029*     |
>
> Similar to Table 1, we see that our model is the best consistent model whereas ProFITi is the overall best for predicting joint distributions.
> ### W5. Worse CRPS and MSE
> Our model achieves the best results among marginalization-consistent models and is overall the second-best model. On the other hand, the best-performing model, ProFITi, has worse CRPS and MSE scores compared to our model when evaluated solely on the univariate marginals (ProFITi-marg). This is because ProFITi is not marginalization-consistent.
> ### W6. Motivation for section C.3 in the appendix
> We included this experiment to compare our model with other marginalization-consistent models studied. Currently, very few models, such as Probabilistic Circuits, are both tractable and marginalization-consistent. These models are designed exclusively for density estimation tasks over a fixed number of variables. Therefore, we aim to compare the performance of our model, which can handle a variable number of inputs, against these models. We have clarified this in the revised version.
>
> [1] Guo, Yuxue, et al. "AI‐based ensemble flood forecasts and its implementation in multi‐objective robust optimization operation for reservoir flood control." Water Resources Research 60.5 (2024): e2023WR035693.

---

### Official Review · Reviewer_XUJD · 2024-11-04

**Soundness:** 2
**Presentation:** 2
**Contribution:** 2
**Rating:** 5
**Confidence:** 5

**Summary:**

The paper considers the problem of forecasting joint distributions of targets in irregular time series, which is a relatively seldom research topic in deep learning community. In particular, the authors show that one of the competitor methods, ProFITi, suffers from marginalization inconsistency. To tackle this issue, the paper proposes a novel method, called moses, that mixes several normalizing flows with Gaussian Processes as source distributions and a separable invertible transformation, aiming to combine the expressivity of normalizing flows with the marginalization consistency of Gaussians. The method was checked in a few experimental settings.

**Strengths:**

The paper has a few strengths overall, which I will outline below.


**Strengths:**
1. Considering the issue of marginalization consistency from the theoretical perspective.
2. Finding the obvious drawbacks (e.g., lack of marginalization consistency) of the baselines.
3. Proposing a simple, but effective, novel method for the considered problem. The model itself is simple and is combining Gaussian processes with a mixture of normalizing flows.

**Weaknesses:**

Despite its strengths, the paper has a few weaknesses.

**Weaknesses:**

1. However, I like the theoretical findings of the proposed method (marginalization consistency), I’m really concerned about the experimental setting and obtained results. In particular, the experimental settings seems to be relatively simple. Moreover, the results are usually worse than ProFITi.
2. Moreover, I don’t know why the authors used the specific normalizing flows (discrete-time) where we need to choose the specific invertible transformations. Why not to use the continuous-time normalizing flows (e.g., FFJORD)? I found the lack of the comparison between the used normalizing flow models a weakness of this paper.
3. Moreover, we don’t know what is the computational (and memory) cost of using the normalizing flows.
4. The comparison in Fig. 1 seems to be unfair to me, since the number of GMM’s parameters is much smaller than the number of moses’s parameters.
5. Regarding the measure of marginalization inconsistency, the authors use Wasserstein distance. However, the number of samples used for obtaining the results is small.
6. Severe computational cost with increasing the number of variables and using of computational heavy normalizing flows are a key limitations of this method. Because of that, I would like to see any higher-dimensional problem to see if this model is feasible to such problems.

**Questions:**

I would like to see especially the experiments and responses to the issues mentioned as weaknesses.

---

> ### Author Response · Authors · 2024-11-25
>
> Thanks for providing detailed feedback.
> ### W1.1: Simple Experimental Settings
> Our experimental setup for probabilistic forecasting of irregular time series with missing values aligns with prior works, including ProFITi, GRU-ODE-Bayes, and Neural Flows, to ensure fair comparisons. In the revised version, we also included an experiment on varying observation and forecasting lengths for further analysis (see revised Supplementary Section C.4). Additionally, since there are marginalization-consistent models for density estimation with a fixed number of variables, we have already compared our approach with other such models in Supplementary Section C.6. of revised version (C.4. of original version).
> ### W1.2: Worse results than ProFITi
> While our model, moses, has slightly worse results compared to ProFITi in predicting joint distributions - $\sim$5% worse likelihoods (note: in terms of actual density, not NLL; see table 1) - moses’ predictions for univariate marginal distributions are significantly better than ProFITi (~51%, see table 2) which is a consequence of ProFITi not adhering to marginalization consistency (fig. 4,5).
>
> In principle, one could train profiti w.r.t to marginals. However, there are exponentially many different subsets of variables one could marginalize with respect to, so this approach is infeasible if we are interested in building a model that performs well for arbitrary queries.
>
> Table 2 demonstrates that moses, trained for the joint NLL, makes very good predictions for the marginal variables, despite only being trained for the joint NLL, which is a direct consequence of the marginalization consistency.
>
> ### W2: Comparison with other Normalizing Flows (e.g. Continuous Normalizing Flows, FFJORD)
> Since the flow needs to be separable, i.e. applied component-wise, we need an expressive univariate flow. We opted for Linear Rational Splines due to their computational efficiency and improved performance over other univariate flows. Dolatabadi et al. (2020) compared FFJORD to spline functions and showed that spline functions provide better density estimates (table 1,2 of Dolatabadi et al. (2020)), and avoids having to deal with numerical ODE solvers.
>
> ### W4: Unfair comparison in fig. 1
> We agree with the reviewer’s concerns, in the experiment the GMM used 7 parameters per mixture component, whereas moses used 133 (7 + 126 for the NF). Currently, we are rerunning the experiment using a different setup, testing the scalability of the approach with respect to the input space dimensionality, under the hypothesis that GMMs do not scale well to high dimensional distributions. We will report the results the earliest possible.
> ### W5: Number of samples for Wasserstein Distance
> We used 1,000 samples for the univariate marginals, which is larger than the sample sizes used in many other studies (e.g., [1], [2]), where sample sizes for evaluation are typically 100. However, we verified that the results are reliable by running 5 repeats per fold. As shown in the table below, the variance between repeats is negligible, showing that N=1000 is sufficient for convergence.
> | Dataset       | 0               | 1               | 2               | 3               | 4               |
> |---------------|-----------------|-----------------|-----------------|-----------------|-----------------|
> | ushcn         | 0.1182 ± 0.0020 | 0.1130 ± 0.0024 | 0.1210 ± 0.0019 | 0.1199 ± 0.0015 | 0.1266 ± 0.0033 |
> | physionet2012 | 0.1308 ± 0.0002 | 0.1009 ± 0.0002 | 0.1239 ± 0.0002 | 0.1152 ± 0.0004 | 0.1194 ± 0.0002 |
> | mimiciii      | 0.1547 ± 0.0003 | 0.1497 ± 0.0004 | 0.1467 ± 0.0003 | 0.1585 ± 0.0005 | 0.1292 ± 0.0006 |
> | mimiciv       | 0.1127 ± 0.0002 | 0.0972 ± 0.0002 | 0.0988 ± 0.0003 | 0.1030 ± 0.0001 | 0.0898 ± 0.0001 |
>
> ### W3,W6: Limitation of Computational Complexity, solving higher dimensional problem
> The normalizing flows used by us are actually computationally light: since they are separable, their Jacobian is diagonal, and hence they can be computed in $𝓞(K)$. The actual bottleneck is computing the density of the Gaussian base distributions, which generally costs $𝓞(K^3)$, as stated in the limitations section.
>
> However, this can be reduced significantly: since we parametrize $Σ=𝕀+UUᵀ$ (eq. 13b), one can actually compute the density in $𝓞(M^2K)$ using the Woodbury and Weinstein–Aronszajn identities. Consequently, our model’s complexity is $𝓞(M^2⋅K⋅D)$, where $K$ is query size, $M$ the embedding dimensionality, and $D$ the number of mixture components, showing that the model is quite scalable when limiting $M$ and $D$.
>
> We forgot to mention this, as in our experiments, we do not use this technique since $K$ is relatively small. We revised the limitations section accordingly.
>
> [1] Rasul, Kashif, et al. "Multivariate Probabilistic Time Series Forecasting via Conditioned Normalizing Flows." ICLR’21.
>
> [2] Zheng, Zhihao, et al. "Better Batch for Deep Probabilistic Time Series Forecasting." AISTATS’24.

---

> > ### Comment · Reviewer_XUJD · 2024-11-28
> > **Thank you for your response**
> >
> > I want to thank the Authors for the detailed response to my questions and found weaknesses. I'm looking forward to see the promised experiment (W4). Overall, I think that added experiments and discussions on the reviewers' questions would improve this paper significantly, I agree with the concerns of Reviewer 9wkz. I think that the overall computational overhead (comparing to baseline) doesn't give us the expected forecasting boost. Regarding the experiment in fig. 1 and W4 (from the authors response), the proposed model has much more parameters than other methods.
> > Finally, I think that this paper still needs to be improved (e.g., by an extensive ablation study of computational complexity, needed wall time, etc. comparing to other methods) and in a current form seems to have insignificant novelty. Moreover, I will raise my score to 5, since the revised version of this paper (**including the promised experiments**) is better than originally. However, I think that the paper should be rejected at this moment.

---

> > > ### Author Response · Authors · 2024-12-04
> > >
> > > Thanks for your time in reviewing the revised version and further comments on our work.
> > >
> > > Firstly, sorry we could not manage to complete the promised experiment. We will add this to the revised version once we have the results.
> > >
> > > Furthermore, the number of parameters used by moses are not significantly high compared to gaussian mixture model (GMM). The major share of the parameters are taken by the encoder to create the covariance matrix compared to the flows for forecasting tasks. For example, for mimiciii dataset we have a total number of 399K parameters of which only 24K are used by flows and remaining for the mixture model and covariance matrix which is common for GMM as well. With slight amount of 24K increase in parameters, moses have significantly better likelihoods compared to GMM.
> > >
> > > Regarding the run times, we are indeed slightly slower than GMM due to additional flows. However, we are quite faster than ProFITi. We provide the run time per epoch for ProFITi and moses for the experiment in Table 1:
> > >
> > > Run time comparison
> > > | Model     | USHCN | PhysioNet 2012 | MIMIC-III | MIMIC-IV |
> > > |-----------|-------|----------------|-----------|----------|
> > > | GMM       | 1s    | 6s             | 19s       | 21s      |
> > > | profiti  | 4s    | 40s            | 67s       | 70s      |
> > > | moses      | 2s    | 27s            | 25s       | 29s      |
> > >
> > > We hope this clarifies the doubts regarding the parameters and run times for moses compared to profiti and GMM

---

### Note · Authors · 2025-09-25

I have read and agree with the venue's withdrawal policy on behalf of myself and my co-authors.

---

### Meta-Review · Area_Chair_YisN · 2024-12-19

**Metareview:**

The paper introduces "moses," a probabilistic forecasting model for irregular time series that addresses marginalization inconsistency by combining normalizing flows with Gaussian Processes. It claims to achieve marginalization consistency and competitive forecasting performance.

Strengths include addressing a neglected area in deep learning, theoretically examining marginalization consistency, and proposing an innovative model structure.

Weaknesses include overall computational overhead without expected performance boost, potential parameter inflation, lack of ablation study, and unclear benefits of marginalization consistency.

The paper is rejected due to concerns about computational without expected performance boost.

**Additional Comments On Reviewer Discussion:**

Reviewer XUJD and Reviewer 9wkz raised the problem that moses suffers from a computational overhead without an expected performance boost, which is not fully solved by the current version.

---

### Decision · Program_Chairs · 2025-01-22

Reject